# SAM-DNMT3A, a strategy for induction of genome-wide DNA methylation, identifies DNA methylation as a vulnerability in ER-positive breast cancers

Mahnaz Hosseinpour [1,2,3], Xinqi Xi[1], Ling Liu [1], Luis Malaver-Ortega[4], Laura Perlaza-Jimenez[1], Jihoon E. Joo[3], Harrison M. York [5], Jonathan Beesley[6], C. Elizabeth Caldon [7], Pierre-Antoine Dugué [2,8,9], James G. Dowty [9], Senthil Arumugam [5,10], Melissa C. Southey [2,3,8] ✉ & Joseph Rosenbluh [1,4] ✉

DNA methylation is an epigenetic mark that plays a critical role in regulating gene expression. DNA methyltransferase (DNMT) inhibitors, inhibit global DNA methylation and have been a key tool in studies of DNA methylation. A major bottleneck is the lack of tools to induce global DNA methylation. Here, we engineered a CRISPR based approach, that we initially designed, to enable site-specific DNA methylation. Using the synergistic activation mediator (SAM) system, we unexpectedly find that regardless of the targeted sequence any sgRNA induces global genome-wide DNA methylation. We term this method SAM-DNMT3A and show that induction of global DNA methylation is a unique vulnerability in ER-positive breast cancer suggesting a therapeutic approach. Our findings highlight the need of caution when using CRISPR based approaches for inducing DNA methylation and demonstrate a method for global induction of DNA methylation.

DNA methylation is a reversible epigenetic modification that plays an important role in regulating gene expression and maintaining the stability of the genome[1]. DNA methylation at promoters regulates gene expression by preventing transcription factors from binding to DNA and initiating transcription[2]. In contrast, DNA methylation at gene body regions can have complex effects on gene expression, sometimes enhancing it[3–5]. Aberrant DNA methylation patterns have been associated with a wide range of diseases, including cancer, neurological disorders, and developmental disorders[1]. Mendelian-like inheritance

of DNA methylation has been shown in various organisms[6]. We have shown that in some cases heritable DNA methylation is associated with susceptibility to different cancer types[7–9]. Although DNA methylation is a fundamental cellular process that is involved in almost every aspect of life, due to the limited available toolkit to study DNA methylation, we still lack comprehensive understanding of how DNA methylation mediates biological processes.

DNA methylation is induced by DNA methyltransferases (DNMT's) that in humans include *DNMT3A* and *DNMT3B* and is maintained by

[1]Department of Biochemistry and Molecular Biology and Cancer program, Biomedicine Discovery Institute, Monash University, Melbourne, VIC, Australia. [2]Precision Medicine, School of Clinical Sciences at Monash Health, Monash University, Melbourne, VIC, Australia. [3]Department of Clinical Pathology, Melbourne Medical School, The University of Melbourne, Melbourne, VIC, Australia. [4]Functional Genomics Platform, Monash University, Melbourne, VIC, Australia. [5]Department of Anatomy and Developmental Biology, Biomedicine Discovery Institute, Monash University, Melbourne, VIC, Australia. [6]Cancer Program, QIMR Berghofer Medical Research Institute, Brisbane, QLD, Australia. [7]The Kinghorn Cancer Centre, Garvan Institute of Medical Research, Sydney, NSW, Australia. [8]Cancer Epidemiology Division, Cancer Council Victoria, Melbourne, VIC, Australia. [9]Centre for Epidemiology and Biostatistics, Melbourne School of Population and Global Health, The University of Melbourne, Melbourne, VIC, Australia. [10]EMBL Australia, Monash University, Melbourne, VIC, Australia. ✉e-mail: melissa.southey@monash.edu; sefi.rosenbluh@monash.edu

*DNMT1*[10]. The DNA methylation mark is removed by the *TET1-3* enzymes[11]. Small molecule inhibitors of DNMT's[12] (e.g. decitabine and azacytidine) induce global DNA de-methylation and have been used as a powerful tool to study the effects of global DNA de-methylation on normal biological processes such as aging[13], and disease conditions such as cancer[12]. A major gap in the study of DNA methylation is that we currently lack strategies for induction of global genome-wide DNA methylation.

Clustered Regularly Interspaced Short Palindromic Repeats (CRISPR) technology offers the potential for precise induction of DNA methylation at desired sites in the genome. Using dCas9 fused to *DNMT3A* Liu et al. developed a programable site-specific DNA methylation platform[14]. Subsequent studies found that this system induces low levels of DNA methylation that is prone to off-target effects[15]. Pflueger et al. [16] found that adopting the SunTag system[17] allows a more specific induction of DNA methylation with less pervasive off-target effects. Here, we show that the synergistic activation mediator (SAM) system[18] induces the highest levels of DNA methylation and that induction of DNA methylation with the SAM system induces global genome-wide non-specific DNA methylation. Our results identify SAM-DNMT3A as an effective tool for induction of global DNA methylation and for studying the effect of global DNA methylation in normal or disease development.

## Results

### Development of SAM-DNMT3A for induction of DNA methylation

Previous studies that adopted the SunTag system for induction of DNA methylation (SunTag-DNMT3A) demonstrated that it is more specific than dCas9-DNMT3A fusions for induction of site-specific DNA methylation[16]. Since the SAM system has been successfully utilised for development of highly active CRISPR based programable transcription activators[18], and since the SAM system requires less plasmids[19], we developed SAM as an approach for induction of DNA methylation (Fig. 1A). In this approach a lentiviral vector expressing a catalytically inactive Cas9 enzyme (dCas9) is fused to one of the three DNMT enzymes and is stably transduced into a cell of interest. Following blasticidine selection for stably integrated cells, a second lentiviral vector expressing a modified sgRNA with two *PP7* RNA binding loops and *DNMT3A-PP7* fusion protein, with a puromycin selection gene is introduced (Fig. 1A). This strategy enables recruitment of three DNMTs to a desired site (Fig. 1B). The High-resolution melting (HRM) assay measures DNA methylation by quantifying the difference in melting curves, in a genomic location, following bisulfite conversion[20]. To test the ability of this approach to induce site-specific DNA methylation, we designed two sgRNAs targeting CpG sequences at the *BRCA1* promoter (Fig. 1C). HEK293T cells were transfected with a non-targeting control (*sgGFP*) or two *BRCA1* targeting sgRNAs fused to either *DNMT3A*, *DNMT3B* or *DNMT1*. Using the HRM assay with *BRCA1* specific primers, we analysed DNA methylation at the *BRCA1* locus, 3 days post-transfection with the indicated DNMTs (Fig. 1D–F and Supplementary Fig. 1A–F). The HRM signal was normalised to control methylated DNA (Supplementary Fig. 1A–F). Compared to un-transfected cells, even the non-targeting control sgRNAs (*sgGFP*) showed an increase in DNA methylation (Fig. 1D–F). We found that the SAM system induced comparable methylation levels at *BRCA1* target sites with all three DNMT's. To compare between SunTag and SAM systems we used the same *BRCA1* targeting sgRNAs with SunTag system fused to *DNMT3A* (Fig. 1G and Supplementary Fig. 1G, H). We found that the SAM system induced higher levels of DNA methylation. All SAM-DNMT systems we tested showed a comparable level of DNA methylation induction, for further experiments we selected the SAM-DNMT3A since it has been used in previous studies[14,16].

### SAM-DNMT3A pooled screens

Pooled genetic screens enable high throughput investigation of genetic perturbations and how they regulate biological processes[21–24].

We have previously reported that some methylation sites are heritable and some of these heritable methylation sites are associated with increased risk of breast[7,8] and prostate[8,9] cancer. Increased cell proliferation is a hallmark of cancer[25] and is associated with many cancer risk-associated genes[24]. To identify if any of these heritable methylation marks induce a cancer-related phenotype, we designed a pooled sgRNA library that targets the top 1000 heritable methylation marks and used this library to identify sites in the human genome that upon methylation induce a proliferation phenotype (Supplementary Data 1).

Since SAM-DNMT3A is directed to the gene promoter it is possible that the observed effect will be related to inhibition of transcription (CRISPRi) rather than induction of DNA methylation. To avoid possible CRISPRi effects of SAM-DNMT3A as a control we did the same screen using a catalytically inactive DNMT3A (SAM-DNMT3A-inactive). Using the HRM assay on cells transfected with SAM-DNMT3A or SAM-DNMT3A-inactive with an sgRNA targeting the *BRCA1* promoter, we found that consistent with previous reports[16], active DNMT3A is required for induction of DNA methylation (Supplementary Fig. 2). Our sgRNA library includes 10,286 sgRNA targeting 1009 genomic regions in the human genome (Supplementary Data 1). For each of these regions, we designed 10 sgRNAs located at the centre of the identified DNA methylation peak (Supplementary Data 1). In addition, we included 737 negative control sgRNAs targeting non-human genes or 262 sgRNAs targeting the *AAVS1* region. MCF7, T47D or BRE80-T5 cells expressing SAM-DNMT3A or SAM-DNMT3A-inactive were transduced with a lentiviral sgRNA library at a multiplicity of infection (MOI) of 0.3, to ensure one sgRNA/cell. Following selection cells were cultured for 21 days and DNA extracted from these cells was used for quantification of sgRNA abundance (Fig. 2A). The effect of an sgRNA on cell proliferation is calculated by comparing sgRNA abundance in SAM-DNMT3A and SAM-DNMT3A-inactive (Supplementary Data 2).

We used this approach to screen three breast cell lines. MCF7 and T47D, two ER-positive breast cancer cell lines, and BRE80-T5, a normal immortalised ER-negative mammary epithelial cell line[26]. By comparing sgRNA abundance in cells with SAM-DNMT3A and SAM-DNMT3A-inactive we found no significant difference (Fig. 2B–D and Supplementary Data 2). Using the MAGeCK algorithm[27] we calculated a gene score for each of these regions and found no significant difference between cells with active or inactive DNMT3A (Fig. 2E–G and Supplementary Data 2). These results suggest that either none of these methylated regions have a functional impact or that site-specific DNA methylation induced with SAM-DNMT3A is non-specific.

### SAM-DNMT3A induces genome-wide global non-specific DNA methylation

We measured the specificity of the SAM-DNMT3A system using the EPIC v1.0 methylation array. We quantified the levels of DNA methylation in BRE80-T5 or T47D cells expressing SAM-DNMT3A or SAM-DNMT3A-inactive. Following transduction with no sgRNA or an sgRNAs targeting *AAVS1*, safe harbour region, or sgRNAs targeting regions in the genome containing hereditable DNA methylation marks[7]. Infected cells were selected with puromycin and 7 days post-transduction, genomic DNA extracted from these cells was used for bisulfite conversion and quantification of global DNA methylation using the Infinium Methylation EPIC v1.0 microarray (Fig. 3A).

For each CpG site in the genome, we calculated the fold change in DNA methylation by comparing DNA methylation in cells expressing SAM-DNMT3A and cells with SAM-DNMT3A-inactive. Using the mean fold change of global DNA methylation across all sites in the human genome we found that compared to cells with no sgRNA cells expressing a targeting sgRNA directed to any region of the genome had a strong effect on induction of DNA methylation throughout the genome (Fig. 3B). In both BRE80-T5 cells and T47D cells DNA methylation was increased compared to the no sgRNA control (Fig. 3B). The increase in DNA methylation was observed throughout the genome

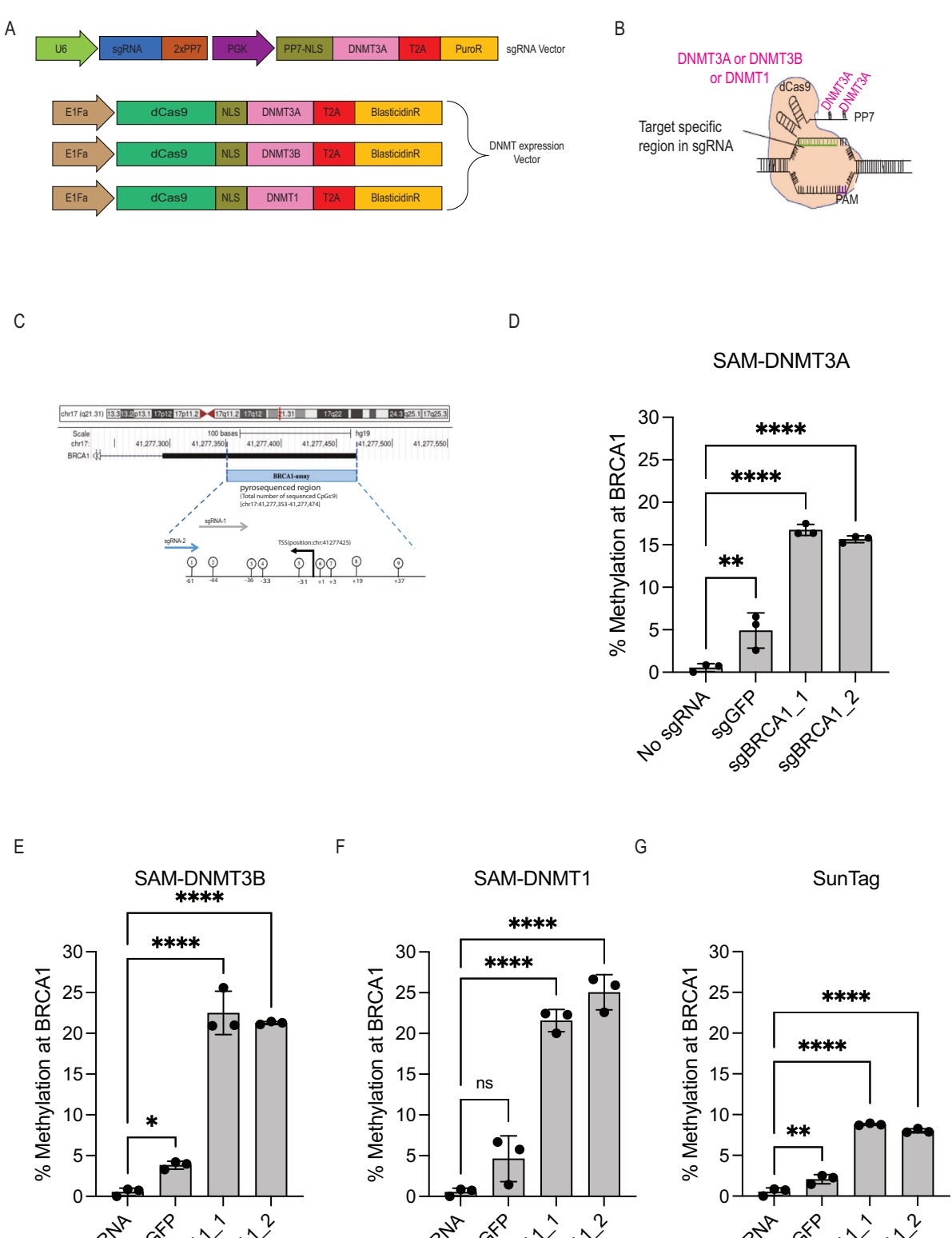

and was not restricted to a specific chromosome or region (Fig. 3C, D and Supplementary Fig. 3A, B).

To further validate the effect of SAM-DNMT3A on global DNA methylation, we repeated this experiment in two additional cell lines with three biological replicates for every condition. MCF7 or BT549 cells were transduced with SAM-DNMT3A or SAM-DNMT3A-inactive with or without an *AAVS1* targeting sgRNA in triplicate and the EPIC

v1.0 array was used to measure global DNA methylation. Multi-dimensional scaling (MDS) analysis showed that in both cell lines replicates clustered together and that cells expressing SAM-DNMT3A clustered away from cells with no sgRNA (mock) or cells expressing SAM-DNMT3A-inactive (Fig. 3E and Supplementary Fig. 3C). For each replicate we calculated the fold change in DNA methylation by comparing DNA methylation levels in empty cells or cells expressing

**Fig. 1 | SAM-DNMT3A induces high levels of DNA methylation at desired sites.**
**A** Vectors used for development of SAM-DNMT3A. The sgRNA vector contains
puromycin resistance and a PP7-DNMT3A fusion protein. The DNMT expression
vector contains a blasticidine resistance gene and a dCas9 enzyme fused to a DNMT
protein. **B** The SAM-DNMT system. At each target site three DNMT enzymes are
recruited. **C** sgRNAs used for induction of DNA methylation at the *BRCA1* promoter.
**D** DNA methylation measured using HRM assay at *BRCA1* promoter 3 days following
transfection of HEK293T cells with the SAM-DNMT3A system. *p*-value was calcu-
lated using one-way ANOVA. Data are mean ± SD, *n* = 3 biological replicates.
($P$ = 3.69e-03 (*sgGFP*), 2.94e-07 (*sgBRCA1_1*), 5.06e-07 (*sgBRCA1_2*)). **E** DNA methy-
lation measured using HRM assay at *BRCA1* promoter 3 days post-transfection of

HEK293T cells with the SAM-DNMT3B system. *p*-value was calculated using one-way
ANOVA. Data are mean ± SD, *n* = 3 biological replicates. ($P$ = 4.68e-02 (*sgGFP*), 1.32e-
07 (*sgBRCA1_1*), 2.07e-07 (*sgBRCA1_2*)). **F** DNA methylation measured using HRM
assay at *BRCA1* promoter 3 days post-transfection of HEK293T cells with the SAM-
DNMT1 system. *p*-value was calculated using one-way ANOVA. Data are mean ± SD,
*n* = 3 biological replicates. ($P$ = 7.35e-02 (*sgGFP*), 2.35e-06 (*sgBRCA1_1*), 7.24e-07
(*sgBRCA1_2*)). **G** DNA methylation measured using HRM assay at *BRCA1* promoter
3 days post-transfection of HEK293T cells with the SunTag-DNMT3A[16] system. *p*-
value was calculated using one-way ANOVA. Data are mean ± SD, n = 3 biological
replicates. ($P$ = 3.16e-03 (*sgGFP*), 1.31e-08 (*sgBRCA1_1*), 2.88e-08 (*sgBRCA1_2*)). NS,
not significant, *$P$ ≤ 0.05, **$P$ ≤ 0.01, ***$P$ ≤ 0.001, ****$P$ ≤ 0.0001.

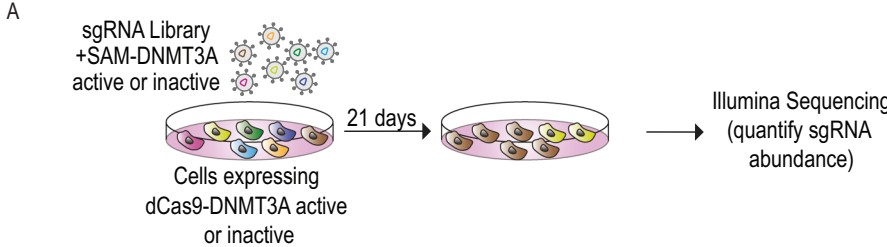

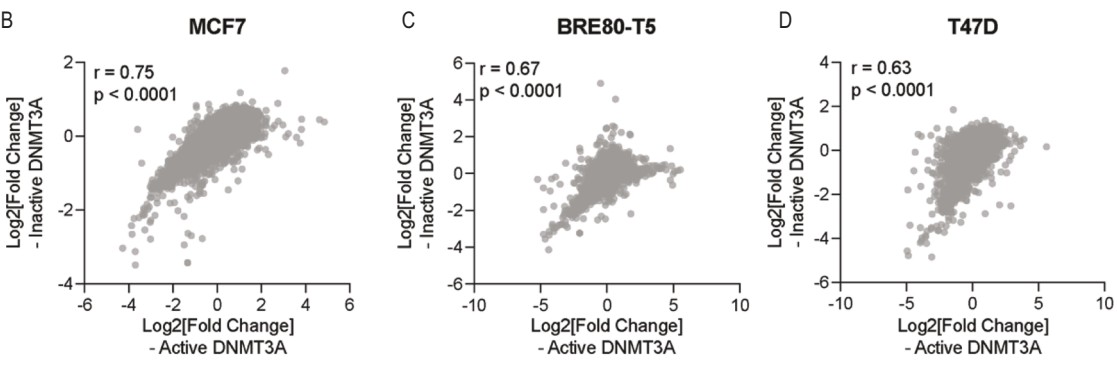

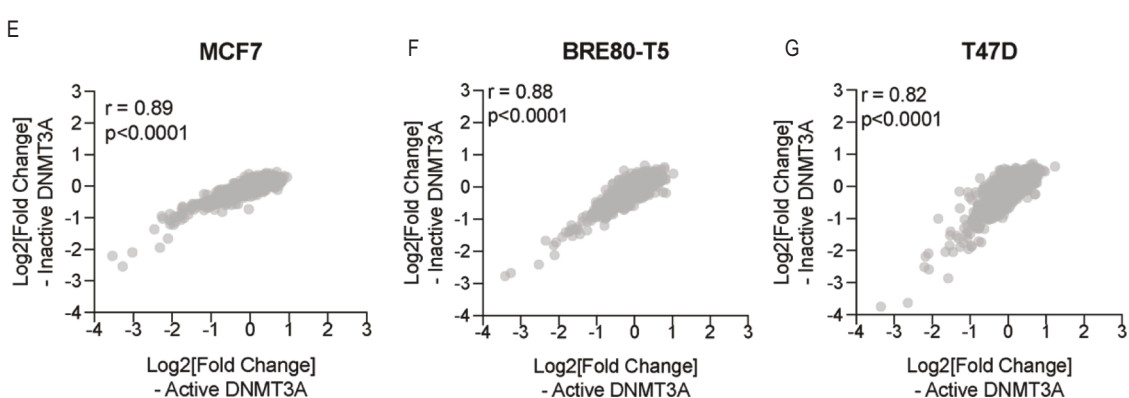

**Fig. 2 | SAM-DNMT3A pooled screens do not identify any hits associated with
proliferation. A** Strategy for pooled DNA methylation screen used to identify DNA
methylation sites that affect cell proliferation. Correlation (Pearson) of sgRNA
abundance between cells with active and inactive DNMT3A. *p*-value calculated
using a two-tailed Pearson correlation analysis. for (**B**) MCF7 ($P$ = 4.34e-56).

**C** BRE80-T5 ($P$ = 5.21e-32). **D** T47D ($P$ = 1.04e-115). Correlation (Pearson) of gene
scores (average of all sgRNAs targeting a methylation region) between cells
expressing active or inactive DNMT3A *p*-value calculated using a two-tailed Pearson
correlation analysis for (E) MCF7 ($P$ = 7.84e-10). **F** BRE80-T5 ($P$ = 8.76e-14). **G** T47D
($P$ = 7.16e-45).

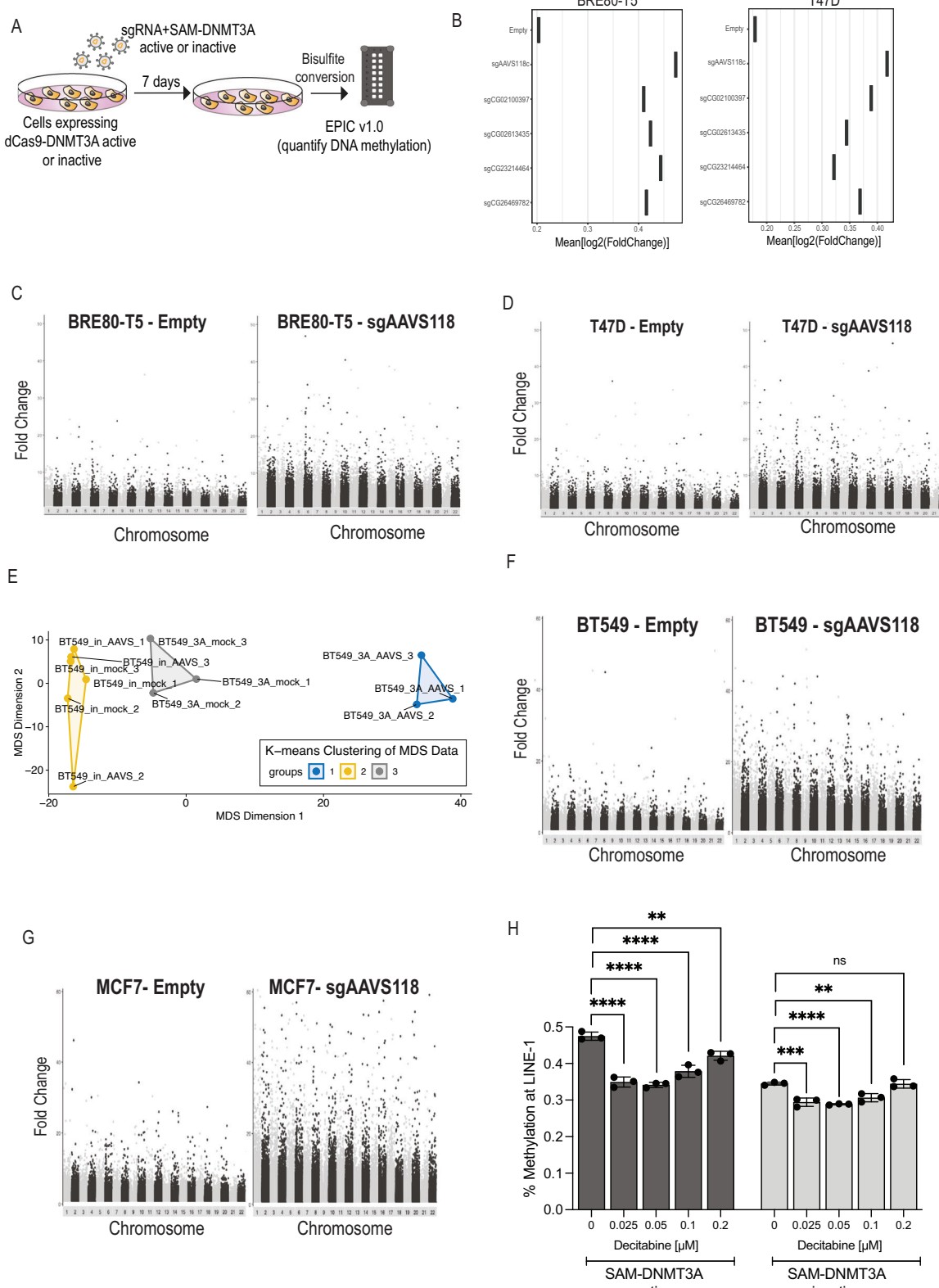

SAM-DNMT3A to cells with SAM-DNMT3A-inactive (Supplementary Data 1). Using the average fold change across three biological replicates at each methylation site we found that SAM-DNMT3A induced high levels of global DNA methylation throughout the genome (Fig. 3F, G).

We used a random sampling approach to evaluate whether observed CpG methylation patterns were enriched in specific regulatory or genomic features. For each CpG site in the EPIC v1 array, we calculated the number of differentially methylated CpGs (defined as Fold Change ≥ 2) in a range of MCF7 genomic annotations, using chromHMM segmentation to define 10 chromatin states[28]. We first intersected EPIC CpG sites with each annotation and counted the number of significantly methylated sites (Fold Change ≥ 2). To generate background expectations, we randomly selected the same

**Fig. 3 | SAM-DNMT3A induces global non-specific DNA methylation.**
**A** Illustration of experiment aimed to identify global methylation changes induced by SAM-DNMT3A. **B** For each methylation site the fold change in methylation was calculated by comparing methylation levels in cells with SAM-DNMT3A and SAM-DNMT3A-inactive. The average fold change across all DNA methylation sites is plotted for each sgRNA. **C** Manhattan plot showing all methylation sites in BRE80-T5 cells with no sgRNAs or with *sgAAVS1_118*. **D** Manhattan plot showing all methylation sites in T47D cells with no sgRNAs or with *sgAAVS1_118*. **E** MDS dimensionality plot using all replicates of EPIC 1.0 array for BT549 cells transduced with mock (control), SAM-DNMT3A-inactive or SAM-DNMT3A-active with *sgAAVS1_118*. **F** Manhattan plot showing all methylation sites in BT549 cells with no

sgRNAs or with *sgAAVS1_118*. Each point is the average of three independent replicates. **G** Manhattan plot showing all methylation sites in MCF7 cells with no sgRNAs or with *sgAAVS1_118*. Each point is the average of three independent replicates. **H** HRM assay using LINE-1 probes in MCF7 cells expressing SAM-DMT3A-active or SAM-DNMT3A-inactive with *sgAAVS1_118* in cells treated for 48 h with 0.2 μM of the DNMT inhibitor decitabine. *p*-value was calculated using one-way ANOVA. Data are mean ± SD, *n* = 3 biological replicates. (*P* = SAM-DNMT3A-active 8.65e-07 (0.025 μM), 4.87e-07 (0.05 μM), 1.01e-05 (0.1 μM), 1.31e-03 (0.2 μM), SAM-DNMT3A-inactive 1.54e-04 (0.025 μM),6.07e-05 (0.05 μM), 1.33e-03 (0.1 μM), 1.00 (0.2 μM)). NS, not significant, *$P ≤ 0.05$, **$P ≤ 0.01$, ***$P ≤ 0.001$, ****$P ≤ 0.0001$.

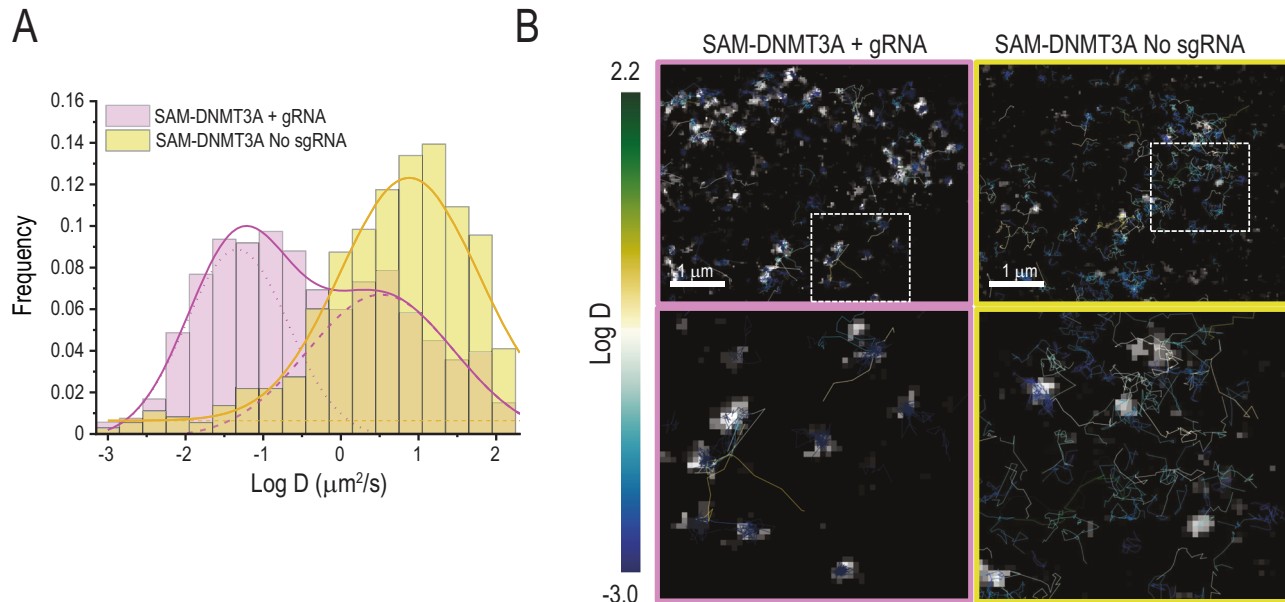

**Fig. 4 | SAM-DNMT3A scans the genome in the presence of an sgRNA.** Dynamics of HaloTag-SAM-DNMT3A measured in HeLa cells transfected with SAM-DNMT3A in the presence or absence an *AAVS1* sgRNA. **A** Quantification of dCas9-DNMT3A movement in the presence or absence of a sgRNA. Data was collected from 534

(SAM-DNMT3A + sgRNA, *n* = 5 cells) and 366 (SAM-DNMT3A No sgRNA, *n* = 3 cells) trajectories. **B** Representative images (from *n* = 5 cells for sgRNA and *n* = 3 cells for No sgRNA control experiments) of trajectory movement of dCas9-DNMT3A in the presence or absence of an sgRNA. Colour bar represents degree of movement.

number of CpG sites from the full array and counted the number of significantly methylated sites (Fold Change ≥ 2). We repeated the random sampling process 100 times to generate a null distribution. We mean centred and scaled the random distributions and calculated Z scores for each observed count, where a Z score > 0 indicates enrichment. We found that DNA methylation was enriched in promoter and heterochromatin regions in both mock and in *sgAAVS1* transduced regions (Supplementary Fig. 3D, E). While *sgAAVS1* showed a global 3-fold increase in DNA methylation, relative enrichment in each annotation was similar in mock and *sgAAVS1* transduced cells. These results suggest that induction of DNA methylation by SAM-DNMT3A is not specific to a particular genomic region and is randomly deposited across the genome.

To confirm that the global methylation induced by SAM-DNMT3A is mediated by DNMT3A enzymatic activity we used the DNMT inhibitor decitabine. MCF7 cells expressing SAM-DNMT3A or SAM-DNMT3A-inactive and an sgRNA targeting *AAVS1* were treated with decitabine. *LINE-1* is a retrotransposon sequence with estimated 500,000 copies in the human genome[29]. To measure global DNA methylation, we used the HRM assay with *LINE-1* probes. We found, that SAM-DNMT3A with an *AAVS1* targeting sgRNA induced an increase in *LINE-1* methylation that was supressed by decitabine (Fig. 3H). Based on these results we conclude that in the presence of an sgRNA the SAM-DNMT3A system induces non-specific global genome DNA methylation that is not related to the sgRNA identity or cell line used.

## Mechanism of SAM-DNMT3A induction of global DNA methylation

In the presence of an sgRNA Cas9 scans the genome in search of a sequence that complements the sgRNA spacer sequence[30]. Since we observed that induction of DNA methylation by SAM-DNMT3A is not sequence specific we hypothesised that dCas9-DNMT3A deposits methylation marks throughout the genome during the process of DNA scanning. To measure the dynamics of SAM-DNMT3A we fused a HaloTag domain to C terminus of dCas9-DNMT3A. HeLa cells were transfected with HaloTag-dCas9-DNMT3A and empty control or an *AAVS1* targeting sgRNA. Following treatment with a cell permeable fluorescent HaloTag ligand dCas9-DNMT3A dynamics were measured with Single Particle tracking using highly inclined and laminated optical sheet microscopy (HiLo)[31]. We found that like Cas9 the mobility of HaloTag-dCas9-DNMT3A was dramatically reduced in the presence of an sgRNA (Fig. 4A). In the presence of an sgRNA, HaloTag-dCas9-DNMT3A was bound throughout the chromosome (Fig. 4A, B and Supplementary Movies 1 and 2) suggesting that the dCas9-DNMT3A sgRNA complex searches the DNA for a target sequence. These observations are consistent with previous reports of Cas9 mechanism of action[30] and suggest that dCas9-DNMT3A induces global DNA methylation in the presence of an sgRNA when it scans the genome for a sequence that matches the spacer.

## Global induction of DNA methylation is a vulnerability in ER-positive breast cancer

During the above-described experiments, we noticed that ER-positive breast cancer cell lines expressing the SAM-DNMT3A system with any sgRNA were difficult to culture for long periods suggesting that induction of global DNA methylation is a vulnerability in ER-positive breast cancers. To test this hypothesis, we measured global DNA methylation levels in ER-positive and ER-negative breast cancer cell lines. We used the HRM assay with probes targeting *LINE-1* sequences. We found that basal levels of DNA methylation were lower in ER-positive breast cancers compared with ER-negative breast cancers (Fig. 5A and Supplementary Fig. 4A, B). These observations are consistent with previous reports using the TCGA dataset showing lower levels of DNA methylation at *ESR1* enhancers in ER-positive breast cancers[32]. We validated these results in an independent cohort of 401 breast cancers with known ER status that we profiled DNA methylation using the HumanMethylation450K (HM450K) BeadChip array[33] and found that ER+ tumours have lower levels of DNA methylation at *ESR1* enhancers (Supplementary Fig. 4C). These observations suggest that low levels of DNA methylation are important in ER-positive breast cancer.

To directly measure the effect of inducing DNA methylation on proliferation of ER-positive breast cancer cells, we used a panel of 7 breast cancer cell lines (4 ER-positive and 3 ER-negative) expressing SAM-DNMT3A or SAM-DNMT3A-inactive with no sgRNA or two *AAVS1* targeting sgRNAs that induce global DNA methylation (Fig. 5B, C). We found that induction of genome-wide DNA methylation had no effect on proliferation in ER-negative breast cancers. Conversely, in ER-positive breast cancer cell lines induction of DNA methylation significantly inhibited cell proliferation (Fig. 5D).

To validate that reduction in proliferation we observed in ER-positive breast cancer cells is due to global induction of DNA methylation we used a rescue experiment. MCF7 (ER-positive) cells expressing SAM-DNMT3A or SAM-DNMT3A-inactive with an *AAVS1* targeting sgRNA were treated with increasing concentrations of the DNMT inhibitor decitabine (Fig. 5E and Supplementary Fig. 4D). High concentrations of decitabine were toxic in all conditions (Supplementary Fig. 4D), however, low concentrations of decitabine rescued growth of MCF7 cells containing the SAM-DNMT3A (Fig. 5E and Supplementary Fig. 4D) demonstrating that SAM-DNMT3A induced proliferation arrest is due to induction of global DNA methylation. These observations demonstrate that SAM-DNMT3A induces global DNA methylation in a cell line independent manner and that induction of DNA methylation is a vulnerability in ER-positive breast cancers.

## Discussion

DNA methylation is the most prevalent epigenetic mark in the human genome. It is a key regulator of normal biological processes and is deregulated in various disease phenotypes[1]. DNA methylation is a fundamental epigenetic modification. It is one of the first epigenetic marks that is induced in the developing stem cell and is critical for lineage commitment[34]. Furthermore, DNA methylation at specific loci could be inherited to off-springs and in some cases is associated with disease risk[7,9]. Global induction or suppression of DNA methylation is a common cellular mechanism to regulate gene expression and is mediated by DNMT and TET enzymes[1]. Despite the clear importance of understanding how DNA methylation regulates biological processes our current toolbox is limited to DNMT inhibitors. Specifically, current tools to study DNA methylation include strategies for profiling of DNA methylation in cells and tissues and methods to perturb DNA methylation and assess the phenotypic outcomes. The global DNA methylation status in the human genome is measured using bisulfite conversion linked to microarrays or sequencing platforms[35]. For functional studies of DNA methylation DNMT inhibitors are commonly used to induce global DNA de-methylation[12]. A major gap is the lack of

tools to induce global DNA methylation. Here, we show that SAM-DNMT3A is a simple robust and effective strategy for induction of global DNA methylation. In the presence of an sgRNA, SAM-DNMT3A induces rapid high-level, genome-wide DNA methylation. SAM-DNMT3A is an important tool for studies of how DNA methylation is associated with phenotypes in normal or disease conditions.

Our results show that SAM-DNMT3A induces genome-wide methylation. Using functional pooled screens we found no consistent phenotypes with SAM-DNMT3A and in global methylation profiling studies we found that any sgRNA we used induced global methylation throughout the genome. Nevertheless, our study finds that CRISPR based systems for induction of DNA methylation are highly prone to off-target effects. Previous studies suggested that systems like SunTag induce higher levels of DNA methylation and are less affected by off-target methylation events[16]. Furthermore, Nunez et al et al.[36] showed that a transient CRISPR based system induces a highly specific and heritable DNA methylation mark. These reports are in contrast with our findings. It is possible that these approaches observe a more specific effect since they use a transient expression system that is different from our system which uses systemic expression of SAM-DNMT3A.

We show that SAM-DNMT3A induced global DNA methylation occurs when the dCas9-DNMT3A sgRNA complex scans the genome to search for its target. By measuring the dynamics of SAM-DNMT3A in the presence or absence of an sgRNA, we found that like Cas9[30], SAM-DNMT3A in complex with an sgRNA scans the genome in search of a spacer target. Together with our results showing that global DNA methylation induced by SAM-DNMT3A is not specific to a genomic region and is randomly distributed throughout the genome, these results suggest that SAM-DNMT3A induces global DNA methylation during the process of scanning the genome. Since the same mechanism would also apply for other CRISPR based approaches, our results suggest that studies using CRISPR based methylation to silence genes, need to take care and ensure that observed phenotypes are not mediated by an off-target global DNA methylation event. Using designer proteins that could only bind to Cas9 once it recognises its target sequence may be able to be used for induction of site-specific DNA methylation.

Previous studies showed that low levels of DNA methylation at ER enhancers is hallmark of ER-positive breast cancers in 2D cultures as well as patient tumours[32]. Here, using SAM-DNMT3A, we show, that low levels of DNA methylation are required for proliferation in ER-positive breast cancers and that induction of DNA methylation is a unique vulnerability in these cancers. Importantly, induction of DNA methylation had no effect on proliferation of ER-negative breast cancer cell lines or on most other cell lines tested with this system. Although currently, small molecules that induce global DNA methylation are not available, these observations suggest that strategies to induce global DNA methylation would be valuable as a treatment in ER-positive breast cancers. These findings could also have implications to ER-positive cancers that are resistant to ER inhibitors. Previous reports suggest that ER-positive breast cancers that are resistant to endocrine therapy have higher levels of DNA methylation[37,38]. Based on our findings showing ER-positive breast cancers require low levels of DNA methylation suggest that ER dependency and DNA methylation are directly connected and that it is possible that escapers of this process are no longer sensitive to endocrine therapy or to DNA methylation induction. Our work provides a method for induction of global DNA methylation and suggests that caution should be taken when using CRISPR based approaches to induce site-specific DNA methylation.

## Methods

### Cell lines and media

Cell-line identity was verified by STR at the Australian Genomics Research Facility (AGRF). MCF7 and T47D cells were a gift from

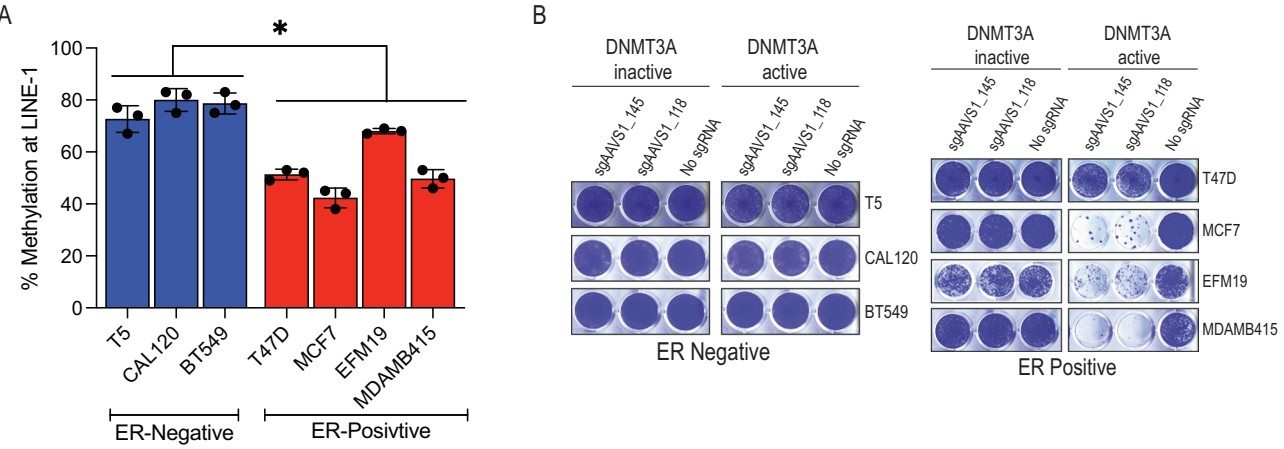

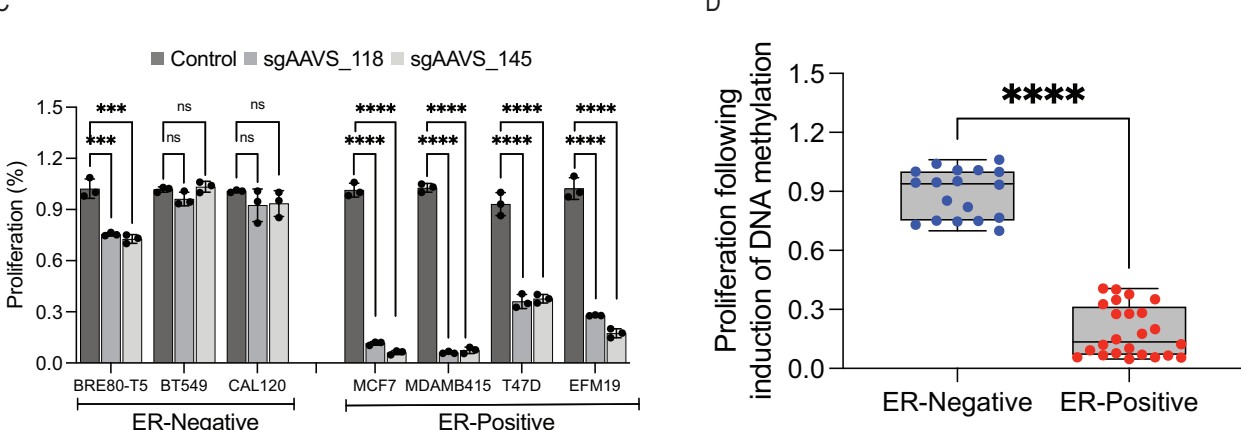

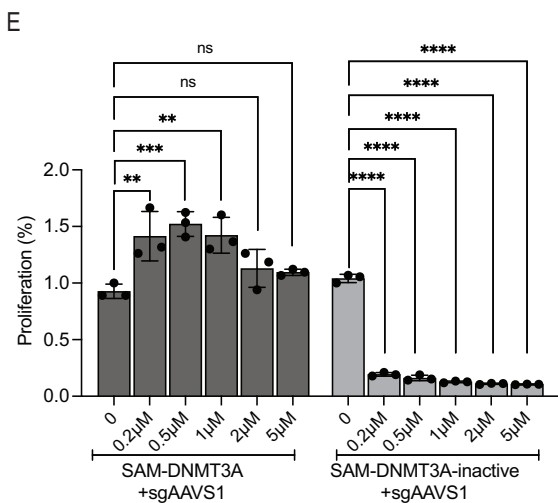

Prof. Georgia Chenevix-Trench (QIMR-Berghofer). HEK293FT cells were from Thermo-Fisher (#R70007). All other cell lines used in this study were from ATCC. HEK293FT cell lines were maintained in the complete DMEM medium consisting of Dulbecco's Modified Eagle Medium (DMEM; Thermo-Fisher Scientific 11965092), 1% Penicillin-Streptomycin (Pen-Strep; Sigma-Aldrich, P4333), 10% Fetal Bovine Serum (FBS; Bovogen, SFBS) and 1% L-glutamine (Sigma-Aldrich, G7513), and supplemented with 1% MEM™ Sodium Pyruvate (Thermo-Fisher

Scientific, 11360070) and 1% MEM™ Non-Essential™ Amino Acids (Thermo-Fisher Scientific, 11140050). MDAMB415 cell lines were cultured in the complete DMEM medium with addition of 0.01 mg/ml insulin from bovine pancreas (Sigma-Aldrich, 15500). K5+/K19- were a gift from Prof. Vilma Band (University of Nebraska). K5+/K19- is a human mammary epithelial cells (HMECs) that have been immortalised with human telomerase reverse transcriptase (TERT)[26] and were maintained in DFCI media[24]. BRE80-T5 was a gift from Prof. Roger

**Fig. 5 | Induction of DNA methylation is a vulnerability in ER positive breast cancers. A** DNA methylation measured in ER-positive and ER-negative breast cells using HRM assay at *LINE-1* sequences following transduction of SAM-DNMT3A with *sgAAVS1_118*. Data are mean ± SD, *n* = 3 biological replicates. *p*-value was calculated using unpaired two-tailed T.test (*P* = 0.015). **B** Images of Crystal violet staining of ER positive and ER negative breast cancer cell lines containing active or inactive DNMT3A 7 days post-transduction with no sgRNA or *AAVS1* targeting sgRNAs. **C** Proliferation changes of all *AAVS1* sgRNAs in ER-positive and ER-negative breast cancers. *p*-value was calculated using one-way ANOVA. Data are mean ± SD, *n* = 3 biological replicates. (*P* = 2.00e-04, 1.00e-04 (BRE80-T5), 0.11, 0.76 (BT549), 0.32, 0.40 (CAL120), 1.77e-08, 1.25e-08 (MCF7), 1.7e-09, 1.9e-09 (MDAMB415), 1.19e-05,

1.41e-05 (T47D), 8.45e-07, 3.88e-07 (EFM19)). **D** Proliferation changes (from (**C**)) in ER-positive and ER-negative breast cancer cell lines following transduction with SAM-DNMT3A. Each dot is an average of three biological replicates. *p*-value was calculated using unpaired two-tailed *T*-test (*P* = 5.05e-20). **E** Proliferation of MCF7 cells with SAM-DNMT3A or SAM-DNMT3A-inactive and an *AAVS1* targeting sgRNA (*sgAAVS1_118*) treated with the indicated concentrations of the DNMT inhibitor decitabine. Data are mean ± SD, *n* = 3 biological replicates. *p*-value was calculated using one-way ANOVA. Data are mean ± SD, *n* = 3 biological replicates. (*P* = SAM-DNMT3A 4.5e-03 (0.2 µM), 9e-04 (0.5 µM), 4e-03 (1 µM), 0.31 (2 µM), 0.47 (5 µM), SAM-DNMT3A-inactive 4e-15 (0.2 µM), 2e-15 (0.5 µM), 1e-15 (1 µM), 1e-15 (2 µM), 1e-15 (5 µM)).

Reddel (University of Sydney). BRE80-T5 is an immortalised mammary epithelial cell line[39] was maintained in the complete RPMI-1640 medium (Thermo-Fisher Scientific #11875093) containing 10% FBS, 1% L-glutamine, 1% Pen-Strep. BT549 and EFM19 cells were maintained in the complete RPMI-1640 medium (Thermo-Fisher Scientific #11875093) containing 10% FBS, 1% L-glutamine, 1% Pen-Strep. MCF7 and T47D cells were cultured in the complete RPMI-1640 medium (Thermo-Fisher Scientific #11875093) supplemented with 0.01 mg/ml insulin and 2.5% HEPES (Thermo-Fisher Scientific #15630080). CAL120 cell lines were cultures in the complete DMEM medium. All cell lines were kept at 37°C in an incubator containing 5% $CO_2$.

## Quantification of global DNA methylation using EPIC array
The Infinium Human Methylation EPIC BeadChip (Illumina, USA) array was performed according to manufacturer's instructions. To quantify the genome-wide methylation status of bisulfite-converted DNAs. The raw intensity signals obtained from scanning of Bead-Chips in EPIC were processed with R using the *minfi* package[40]. Methylation CpG (5'-C-phosphate-G-3') islands were annotated against the genome using the *IlluminaHumanMethylationEPI-Canno.ilm10b2.hg19* package. The quality of the methylated positions was assessed using the R function *detectionP* from *minfi*[40] and low-quality positions were discarded. Low-quality positions were detected when the methylated and unmethylated channel contained background signal levels (*p-value > 0.01*). Data normalization was performed using the R function *preprocessQuantile* from *minfi* and beta-values were generated with the R function *getBeta* from *minfi*. We calculated the fold-change in methylation for each CpG site by comparing the beta-values per CpG between SAM-DNMT3A and cells with SAM-DNMT3A-inactive. Manhattan plots were generated using ggplot2 for each comparison.

## Plasmid constructions
The following plasmids were obtained from Addgene; Plenti-EF1a-SPdCas9-DNMT3B-2A-Blast (dCas9-DNMT3B) (Addgene#71217), pCCL-PGK-SPdCas9-BFP-DNMT1(dCas9-DNMT1) (Addgene#66818) pHRdSV40-NLS-dCas9-24×GCN4-v4-NLS-P2A-BFP-dWPRE (Addgene#60910), pEF1a-NLS-scFvGCN4-DNMT3a (Addgene#100941), pXPR502 (Addgene # 96923), pLenti-EF1a-SPdCas9-DNMT3A-2A-Blast (dCas9-DNMT3A; Addgene#71216), pLenti-EF1a-SPdCas9-DNMT3A(E752A)-2A-Blast (dCas9-inactiveDNMT3A; Addgene# 71218) and pLentiGuide-Puro (Addgene#52963).

For generation of the SAM-DNMT3A sgRNA vector. The transcriptional activator of p65-HSF1 was removed by *KpnI* digestion from pXPR502. The DNMT3A sequence was PCR amplified (PCR primers in Supplementary Data S3) from dCas9-DNMT3A and ligated using gibson ligation. To enable sgRNA cloning via the *ESP3I* restriction site we removed the CGTCTC from the DNMT3A sequence. Using PCR based mutagenesis we generated a point mutation CGTGTC. This point mutation (c.798 C > G) in DNMT3A removes the *ESP3I* restriction site without altering the amino acids sequence. To generate the negative control SAM-DNMT3A-inactive sgRNA vector, a gBlock sequence of containing the catalytically inactive DNMT3A inactive was inserted to

the *BstXI* and *PstI* sites of SAM-DNMT3A vector. For cloning of sgRNAs into the sgRNA vectors. For SunTag system sgRNAs were cloned to the *ESP3I* sites of pLentiGuide-Puro (Addgene#52963). For SAM-DNMT3A and SAM-DNMT3A-inactive sgRNAs were amplified by PCR and cloned using golden gate cloning to the ESP3I sites of SAM-DNMT3A or SAM-DNMT3A-inactive vector. All plasmids generated in this study are available through Addgene.

## Cell transfections
HEK293FT cells were transfected with indicated vectors using Lipofectamine3000 (Thermo-Fisher#L3000008). Briefly, 0.5e6 HEK293FT cells were seeded in a six-well plate. 24 h post seeding, 1250 ng of each construct were transfected using a 1:1 ratio of DNA to Lipofectamine reagents.

## Generation of Lentiviral particles
Virus particles were made as previously described[24]. Briefly, HEK293FT cells (6.5e6) were plated in 10 cm dish. After 24 h cells were transfected with 8750 ng of indicated vector and 8750 ng of psPAX2 (Addgene#12260) and 875 ng of pMD2.G (Addgene#12259) in Opti-MEM (Thermo-Fisher#51985091) (final volume 500 µl) and 103 µl of Lipofectamine 3000. Supernatant was collected at 48 h and 72 h post-transfection and 30% of FBS was added. Virus particles were kept at −80 °C.

## Transduction of cells with SAM-DNMT's
To induce DNA methylation using *SAM* system cells were infected with a lentiviral vector expressing dCas9-DNMT3A and a modified sgRNA lentiviral vector (Fig. 1A). 1.2e6 cells were plated in 8 ml media and supplemented with 0.5 µg/ml of polybrene in a 10 cm dish and 2 ml of dCas9-DNMT3A expressing virus was added to the media. 24 h later, media was removed and replaced with a media containing 10 µg/ml of blasticidin (Thermo-Fisher#A1113903). Following selection (7 days), cells were maintained and expanded in a media containing 5 µg/ml of blasticidin for later application. To express an sgRNA in these cells, 3e5 cells/well of dCas9-DNMT3A expressing cells were plated on a 6-well plate. 24 h later media was replaced with 1.5 ml of media containing 0.5 µg/ml of polybrene and 500 µl of modified sgRNA lentiviral vector. Following 24 h, the media was replaced with media containing 5 µg/ml of blasticidin and 2 µg/ml of puromycin. After selection (2 days) puromycin was reduced to 1 µg/ml.

## Generation of pooled sgRNA libraries
Pooled sgRNA libraries were generated as previously described[23]. Briefly, For each of heritable methylation marks, 10 sgRNAs were designed (Supplementary Data S1). Each sgRNA had *ESP3I* cut sites and flanking PCR handles (AGGCACTTGCTCGTACGACGCGTCTCACACCG[20ntsgRNA]GTTTCGAGACGTTAAGGTGCCGGGCCCACAT-3'). PCR amplification was performed using primers (Fwd: 5'-AGGCACTTGCTCGTACGACG-3', Rev: 5'-ATGTGGGCCCGGCACCTTAA-3'). Golden gate cloning was applied to insert the library into the *ESP3I* sites of SAM-DNMT3A or SAM-DNMT3A-inactive. Library representation was verified by NGS sequencing.

## SAM-DNMT3A screen

MCF7, BRE80-T5 and T47D cells stably expressing dCas9-DNMT3A or dCas9-DNMT3A-inactive were transduced with the sgRNA pooled library virus at an MOI of 0.3. 24 h later infected cells were selected using Puromycin (2 µg/ml). Cells were maintained with blasticidin and puromycin throughout the screen, to ensure expressions. 21 days post-infection, DNAs were extracted using NucleoSpin Blood XL (Macherey-Nagel#740950). Samples were prepared for sequencing as previously described[23]. Briefly, sgRNAs were PCR amplified using P5_ARAGON and P7_KERMIT primers (https://portals.broadinstitute.org/gpp/public/resources/protocols). Amplicons were pooled and purified on AMPure beads and sequenced on Illumina HiSeq machine.

## Crystal violet proliferation assay

2e4 cells/well were plated on a 24-welll plate in triplicates and allowed to propagate for 7–10 days. Cells were washed with Dulbecco's phosphate-buffered saline (DPBS; ThermoFisher#14190-144) and 1 ml of 10% formalin was added into each well and incubated for 20 min at room temperature. After removal of formalin, 0.5 ml of 0.5% (w/v) crystal violet (Sigma#C0775-25G) was added. After 20 min, crystal violet was removed, and plates were washed with water. For quantification, 0.5 ml of 10% Acetic acid was added to each well and incubated for 30 min. 100 µl of the dissolved solution was added to a 96-well plate and quantified by measuring the OD at 570 nm.

## Analysis of global DNA methylation using the HRM assay

7 days post-infection with the indicated sgRNAs, genomic DNA was extracted using DNeasy Blood and Tissue Kit (Qiagen#69504) and bisulfite-converted using EZ-DNA Methylation-Gold kit. CpGenome™ Human Methylated & Non-Methylated DNA standard set (Sigma#S8001M) was used to prepare a set of methylation standards (0%, 25%, 50% and 100% methylated DNA standards). The PCR amplification of all bisulfite-converted DNA was performed in a 20 µl reaction volume containing 1 µl of bisulfite-converted DNA, 10 µl of MeltDoctor HRM Mastermix (ThermoFisher#4415440), 1 µl of forward and 1 µl of reverse primers (10 µM) and 7 µl of $H_2O$. We targeted the *LINE-1* repeat region to measure global DNA methylation or the *BRCA1* promoter to quantify site-specific DNA methylation using primers indicated in Supplementary Data 3. HRM was initiated by one cycle at 95 °C for 10 min followed by 40 cycles at 95 °C for 15 s 60 °C for 1 min. The melting process was followed by a denaturing step at 95 °C for 10 sec, annealing at 60 °C for 1 min and by a stepwise increase in temperature from 60 °C to 95 °C, with a rate of 0.1 °C per second. Finally, The HRM results were analysed using the Thermo-Fisher HRM software.

## Single particle tracking and HiLo Microscopy

HeLa Cells were grown on cleaned 25 mm coverslips and were mounted on the Attofluor chamber for live-cell imaging. Microscopy were performed using a 100×1.4 NA objective on an Oxford NanoImager microscope. Movies were acquired with 30 ms and were analysed using a custom MATLAB code. Briefly, particles were detected using a Laplacian of a Gaussian (LoG) operator and tracked using a custom tracking routine built on Trackpy was then used to link objects into complete trajectories with the following set of rules: Particles could only link to a single particle in the subsequent frame or can terminate, no merge and split events were considered since the experimental design excludes these events. Linked trajectories were generated by Linear Assignment Problem (LAP)[41]. The identified single particle trajectories were used to generate MSD plots. The MSD curves were fitted with the following equation: $MSD = \langle R \rangle 2 = 4Dt\alpha + 2\sigma 2$, where $D$ is the diffusion coefficient, $t$ is the lag-time, $\alpha$, the scaling exponent, and $\sigma$ is the measurement error. Diffusion coefficients were calculated from tracks with at least 5 consecutive frames. The quality of fit was restricted to $R^2 < 0.8$[42].

## Statistics and reproducibility

All raw and processed data from screens are available in the supplementary data files. No data were excluded from the analyses. Experiments were not randomized, and investigators were not blinded during experiments and outcome assessment. The experiments in all figures were performed in triplicate except for screens in Fig. 2 that were done in duplicate. Results are shown as mean and standard deviation unless stated otherwise. No statistical methods were used to predetermine sample sizes. Graphs were plotted, and statistical analysis was performed with GRAPHPAD PRISM 10.3.1 (GraphPad, San Diego, CA, USA). Statistical methods that were used and significance is indicated in the figure legend. The number of biological repetitions (n) is stated in each figure legend, and every experiment was performed at least twice. Differences between groups were considered statistically significant when $P < 0.05$. In figures, asterisks stand for: *$P < 0.05$; **$P < 0.01$; ***$P < 0.001$; ****$P < 0.0001$.

## Reporting summary

Further information on research design is available in the Nature Portfolio Reporting Summary linked to this article.

## Data availability

All raw and processed data from EPIC arrays are available through GEO accession number: GSE249125. Raw and processed reads from pooled screens are available in the Supplementary Data files. BedGraph tables that could be opened and explored in IGV[43] of EPIC v1.0 arrays in MCF7 and BT549 containing the fold change in DNA methylation are available at: GSE249125. All plasmids made in this study are available through Addgene. Source data are provided with this paper.

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

## Acknowledgements

This work was supported by an NHMRC Synergy grant to J.R. and M.C.S. (grant number: 2011329). J.R. is supported by a Victoria Cancer Agency fellowship (grant number: MCRF20035). M.C.S is supported by a Level 3 NHMRC Investigator grant (grant number: GNT2017325). We thank the Functional Genomics Platform, and the Genomics and Bioinformatics Platform at Monash University for help with loss of function screens and data analysis. We thank Phenomics Australia for support of pooled screening experiments. We thank Dr. Gavin Knott (Monash University) for helpful discussions. We thank Prof. Georgia Chenevix-Trench (QIMR), Prof. Roger Reddel (University of Sydney) and Prof. Vilma Band (University of Nebraska) for providing cell lines. We thank Dr. Lochlan Fennell (University of the Sunshine Coast) for help with the HRM assay.

## Author contributions

Conceptualization: J.R., M.H. and M.C.S. Methodology: J.R., M.H., X.X. L.L., L.M.O. H.Y., C.E.C. Analysis: J.R., M.H., L.P.J, J.E.J, J.B., C.E.C. P.A.D., J.G.D., S.A. M.C.S. Writing—original draft: J.R., M.C.S, M.H. Writing—review and editing: all authors. Supervision: J.R., L.L., S.A. and M.C.S. Funding acquisition: J.R., M.C.S. The authors read and approved the final manuscript.

## Competing interests

The authors declare no competing interests.
