## [Transparent Peer Review file · Nature Communications]

SAM-DNMT3A, a strategy for induction of genome-wide DNA methylation, identifies DNA methylation as a vulnerability in ER-positive breast cancers

Corresponding Author: Professor Joseph Rosenbluh

Version 0:

Reviewer comments:

Reviewer #1

(Remarks to the Author)

The authors here happened to find that SMA-DNMT3A was able to induce genome-wide DNA methylation and DNA methylation could be a potential therapeutic approach for ER-positive breast cancer. Overall, these findings are interesting and useful, but I still have several major concerns for the authors to address.

1. The present finding is important, while it is highly recommended for the authors to explore more about the underlying mechanisms of SMA-DNMT3A-mediated genome-wide methylation.
2. Although DNMT3A showed highest level of site-specific DNA methylation (Fig 1D), its level of non-specific methylation is lower than those of DNMT3B and DNMT1 as shown in 1E and 1F when sgGFP was used. Considering the author mainly want to explore the genome-wide non-specific methylation effect of DNMT, why the authors did not use DNMT3B and DNMT1?
3. The authors found that low decitabine concentration rescued the growth of MCF7 expressing SAM-DNMT3A to some extent, which was consistent with the authors' proposal. However, the authors should still measure the overall methylation level of the cell under different treatment conditions.

Reviewer #2

(Remarks to the Author)

Hosseinpour et al manuscript entitled 'SAM-DNMT3A, a strategy for induction of genome-wide DNA methylation, identifies DNA methylation as a vulnerability in ER-positive breast cancers' describes a new method SAM-DNMT3A that induces global DNA methylation with a unique vulnerability in ER- positive breast cancer suggesting a therapeutic approach.

However, there are major concerns that need to be addressed with regards to the robustness of the experimental design – lack of replicates, lack of experimental details and no comprehensive data analyses to support the conclusions of the paper.

Major Concerns.

- Fig1 D-F it is stated that BRCA1 targeting sgRNAs induced higher levels of DNA methylation- however untransfected controls are not shown to demonstrate the basal level of DNA methylation to compare the transfected levels. There is no quantification of methylation levels to support that statement that SAM-DNMT3A induced the highest levels of DNA methylation as they all look comparable. Please show untransfected DNA methylation levels and perform at least 3 replicate experiments to allow statistical rigour in the BRCA and PTEN and NF1 experiments (if this was done it was not stated).
- The genome-wide approach uses a sgRNA library containing 10,286 sgRNA targeting 1,009 genomic regions in the human genome and methylation levels assessed using the Infinium Methylation EPIC v2.0 microarray. The methylation arrays appear not to be performed in triplicate (as this was not stated) and is minimum for any robust statistical analyses.
- Importantly there needs to be a link to the methylation array data so that the reviewers can review the data and a QC of this data needs to be presented.
- The genome-wide methylation data in (Fig. 3C,D and Supplementary Fig. 3A,B) is largely non informative and shows a lack of robust data analysis.
- Differences in genome wide methylation should also be represented at least using correlation plots of the methylation B-

value of the replicates / and or a correlation matrix between the SAM-DNMT3A and SAM-DNMT3A-inactive and no sgRNA cells samples to determine the genome-wide changes in DNA methylation.

- In addition the differentially methylated sites should be mapped to determine the extent and position of the off target effects- eg Venn diagram of target and off targets and importantly an analysis of the functional annotations and CpG distribution/density of target and non-targeted regions to define what regions show different target specificity.
- Importantly to assess the DNA methylation induced by SAM-DNMT3A in ER-positive and ER negative breast cancer cells analyses of the EPIC array data should be performed to look at DMRs or DMPs and their functional locations- promoters/enhancer and closest genes etc.
- No mechanism is addressed to explain the off target effects nor the susceptibility of the ER+/ER- response differences observed. For example global DNA methylation levels using EPIC analyses were not presented for the 7 breast cancer cell lines tested- this is not shown in Fig 4 D-E- only for LINE-1 sequences – it is important to show the differences in basal levels and change in DNA methylation induced by SAM-DNMT3A genome-wide?
- In the ER+/ER- cell lines are there ER methylated/expression differences? Are the target genes differentially methylated pre post SAM-DNMT3A and SAM-DNMT3A-inactive and no sgRNA cells samples ?
- Finally, but critically, it is stated that EPICv2 was used for the genome-wide methylation data – however in the methods section outlining the quantification of global methylation- what is described are tools used for EPIVv1 analyses and these do not currently work for EPICv2 as it has a different manifest with a subset of different probes- unless the authors only assessed EPICv1 probes- their analyses need further description.

Minor

- DNMTs stand for DNA methyltransferases- not DNA methylases as defined in the paper in the abstract and Introduction. Please modify accordingly.
- In many parts of the manuscript the writing is 'sloppy' , for example
 - o In the introduction last paragraph define the SAM system ie the acronym.
 - o in the first part of the results 'In this approach a lentiviral vector expressing a catalytically inactive Cas9 enzyme (dCas9) fused to a DNMT enzyme, is stably transduced into a cell'- what DNMT enzymes and what cells?

Version 1:

Reviewer comments:

Reviewer #1

(Remarks to the Author)

The authors have added more data to explore the underlying molecular mechanisms, and the revised manuscript has been much improved. I have no more concerns.

Reviewer #3

(Remarks to the Author)

In my opinion, the authors have sufficiently addressed all concerns raised by the reviewer. I congratulate the authors for a study that will be of great use to the epigenetics community, especially those interested in epigenome manipulation tools and DNA methylation in disease.

Reviewer #4

(Remarks to the Author)

I completely agree with the concerns raised by Reviewer #2; however, the authors have adequately addressed them in the rebuttal, which has greatly enhanced the study.

REVIEWER COMMENTS

Reviewer #1 (Remarks to the Author):

The authors here happened to find that SAM-DNMT3A was able to induce genome-wide DNA methylation and DNA methylation could be a potential therapeutic approach for ER-positive breast cancer. Overall, these findings are interesting and useful, but I still have several major concerns for the authors to address.

1. The present finding is important, while it is highly recommended for the authors to explore more about the underlying mechanisms of SAM-DNMT3A-mediated genome-wide methylation.

As requested by the reviewer we have added a new experiment that shows the mechanism of SAM-DNMT3A (Fig. 4 in the revised manuscript). Previous studies have shown that following sgRNA binding, Cas9 scans the DNA in search of the sgRNA target sequence (PMID: 26564855). Since it is very likely that SAM-DNMT3A works in a similar way we used a HaloTag version of SAM-DNMT3A to monitor its dynamics in the presence or absence of an sgRNA. In the revised manuscript we show that like Cas9 also dCas9-DNMT3A scans the DNA in a non-specific way (Fig. 4 in the revised manuscript) in search of the sgRNA target sequence which results in global DNA methylation. This conclusion is further strengthened by our results showing that global DNA methylation induced by SAM-DNMT3A is not specific to a genomic region and is randomly distributed in the genome (Supplementary Fig. 3D,E in the revised manuscript).

2. Although DNMT3A showed highest level of site-specific DNA methylation (Fig 1D), its level of non-specific methylation is lower than those of DNMT3B and DNMT1 as shown in 1E and 1F when sgGFP was used. Considering the author mainly want to explore the genome-wide non-specific methylation effect of DNMT, why the authors did not use DNMT3B and DNMT1?

In the revised manuscript we repeated these experiments with more replicates and controls (Fig. 1D-G and Supplementary Fig. 1A-H in the revised manuscript). We found that the levels of specific and non-specific DNA methylation are higher in the SAM system in comparison to the SunTag system (Compare Fig. 1D-F with Fig. 1G in the revised manuscript). The levels of specific and non-specific DNA methylation were comparable between the different DNA methylases (Fig. 1D-F in the revised manuscript). We chose to use SAM-DNMT3A since DNMT3A has been most widely used for induction of DNA methylation.

3. The authors found that low decitabine concentration rescued the growth of MCF7 expressing SAM-DNMT3A to some extent, which was consistent with the authors' proposal. However, the authors should still measure the overall methylation level of the cell under different treatment conditions.

As requested by the reviewer, we have repeated this experiment at lower decitabine concentrations. The new revised figure shows that SAM-DNMT3A induced global DNA methylation is inhibited at low concentrations of decitabine (Fig. 3H in the revised manuscript).

Reviewer #2 (Remarks to the Author):

Hosseinpour et al manuscript entitled 'SAM-DNMT3A, a strategy for induction of genome-wide DNA methylation, identifies DNA methylation as a vulnerability in ER-positive breast cancers' describes a new method SAM-DNMT3A that induces global DNA methylation with a unique vulnerability in ER- positive breast cancer suggesting a therapeutic approach.

However, there are major concerns that need to be addressed with regards to the robustness of the experimental design – lack of replicates, lack of experimental details and no comprehensive data analyses to support the conclusions of the paper.

Major Concerns.

- *Fig1 D-F it is stated that BRCA1 targeting sgRNAs induced higher levels of DNA methylation-however un-transfected controls are not shown to demonstrate the basal level of DNA methylation to compare the transfected levels.*

We are sorry for not including this in the original manuscript. As requested, we have repeated all these experiments with more replicates using the HRM assay (Fig. 1D-G and Supplementary Fig. 1). The new figure 1 includes a 'no sgRNA' control for all experiments and shows the levels of site specific and non-specific DNA methylation induced by SAM-DNMT's (Fig. 1D-F in the revised manuscript) and by SunTag (Fig. 1G in the revised manuscript).

There is no quantification of methylation levels to support that statement that SAM-DNMT3A induced the highest levels of DNA methylation as they all look comparable. Please show un-transfected DNA methylation levels and perform at least 3 replicate experiments to allow statistical rigour in the BRCA and PTEN and NF1 experiments (if this was done it was not stated).

We have modified Figure 1 and have added for each of the DNMT enzymes 3 replicates. The results are consistent with previous data showing both specific and non-specific DNA methylation is induced by SAM system. We also included in the revised manuscript the unprocessed HRM plots (Supplementary Fig. 1 in the revised manuscript).

- *The genome-wide approach uses a sgRNA library containing 10,286 sgRNA targeting 1,009 genomic regions in the human genome and methylation levels assessed using the Infinium Methylation EPIC v2.0 microarray. The methylation arrays appear not to be performed in triplicate (as this was not stated) and is minimum for any robust statistical analyses.*

We are sorry for not making this point clear. The functional sgRNA screen (Fig. 2) uses a pooled sgRNA library and a phenotypic readout (cell proliferation). In the functional screen we could not measure DNA methylation since it is done in a pooled format (see Fig. 2A for the experimental strategy of this experiment). We modified the text to make this point clear. In Fig. 3 we used EPIC v1.0 array and directly measured the effect of SAM-DNMT3A on induction of DNA methylation. We have added a new experiment that was done in triplicate in two additional cell lines showing that induction of DNA methylation by SAM-DNMT3A is highly reproducible (Fig. 3E and Supplementary Fig. 3C in the revised manuscript).

- *Importantly there needs to be a link to the methylation array data so that the reviewers can review the data and a QC of this data needs to be presented.*

All raw DNA methylation data from EPIC arrays (48 samples) is uploaded to GEO (GEO accession number: GSE249125, reviewer token: ihyfamssnzidlgx). Once the paper is published, we will make this data publicly available.

- *The genome-wide methylation data in (Fig. 3C,D and Supplementary Fig. 3A,B) is largely non informative and shows a lack of robust data analysis.*

We are sorry for not including enough data. In the revised manuscript we have repeated this experiment in triplicate in two different cell lines and we show that SAM-DNMT3A induces a highly robust and reproducible levels of DNA methylation (Fig. 3E and Supplementary Fig. 3C in the revised manuscript). To make the data more accessible we also added as a supplementary data file a bed file that contains the fold change in DNA methylation at every genomic site on the EPIC array. The bed file could be opened and explored using the Integrative Genomic Viewer (IGV - <https://igv.org/doc/desktop/>)(Supplementary Data 1 in the revised manuscript).

- *Differences in genome wide methylation should also be represented at least using correlation plots of the methylation B-value of the replicates / and or a correlation matrix between the SAM-DNMT3A and SAM-DNMT3A-inactive and no sgRNA cells samples to determine the genome-wide changes in DNA methylation.*

We have added a MDS analysis of all individual samples in two cell lines (Fig. 3E and Supplementary Fig. 3C in the revised manuscript) showing that replicate experiments are highly correlated and that SAM-DNMT3A-inactive and empty cells cluster together and cells expressing SAM-DNMT3A cluster separately.

- *In addition, the differentially methylated sites should be mapped to determine the extent and position of the off target effects- eg Venn diagram of target and off targets and importantly an analysis of the functional annotations and CpG distribution/density of target and non-targeted regions to define what regions show different target specificity.*

In the revised manuscript we show that SAM-DNMT3A deposits methylation marks as it scans the DNA in search of a target sequence (Fig. 4 in the revised manuscript). As expected from this mechanism induction of DNA methylation is found throughout the genome and is not restricted to a specific region. To directly address the reviewers comment we used a random sampling approach. Specifically, for each CpG site in the EPIC v1 array, we calculated the number of differentially methylated CpGs (defined as Fold Change ≥ 2) in a range of MCF7 genomic annotations, using chromHMM segmentation to define 10 chromatin states (PMID: 24916973). We first intersected EPIC CpG sites with each annotation and counted the number of significantly methylated sites (Fold Change ≥ 2). To generate background expectation, we randomly selected the same number of CpG sites from the full array and counted the number of significantly methylated sites (Fold Change ≥ 2). We repeated the random sampling process 100 times to generate a null distribution. We mean centred and scaled the random distributions and calculated Z scores for each observed count, where a Z score > 0 indicates enrichment. We found that DNA methylation was enriched in promoter and heterochromatin regions in both mock and in sgAAVS1 transduced regions (Supplementary Fig. 3D,E). While sgAAVS1 showed a global 3-fold increase in DNA methylation, relative enrichment in each annotation was similar in mock and sgAAVS1 transduced cells. These results suggest that induction of DNA methylation by SAM-DNMT3A is not specific to a particular genomic region and is randomly deposited across the genome.

- *Importantly to assess the DNA methylation induced by SAM-DNMT3A in ER-positive and ER negative breast cancer cells analyses of the EPIC array data should be performed to look at DMRs or DMPs and their functional locations- promoters/enhancer and closest genes etc.*

As stated above DNA methylation induced by SAM-DNMT3A is not specific and we did not find any specific genomic region or motif that is differentially methylated in ER-positive cancers (Supplementary Fig. 3D,E). However, we found that the global levels of DNA methylation in ER-positive breast cancer cell lines and tumours is lower than ER-negative cancers (Fig. 5A and Supplementary Fig. 4C in the revised manuscript). This is consistent with previously published results showing low levels of DNA methylation at ER-enhancers (PMID: 26169690). Using SAM-DNMT3A we could show, for the first time, that induction of DNA methylation is a vulnerability in ER-positive cancers.

- *No mechanism is addressed to explain the off-target effects nor the susceptibility of the ER+/ER- response differences observed. For example, global DNA methylation levels using EPIC analyses*

were not presented for the 7 breast cancer cell lines tested- this is not shown in Fig 4 D-E- only for LINE-1 sequences – it is important to show the differences in basal levels and change in DNA methylation induced by SAM-DNMT3A genome-wide?

Low levels of DNA methylation in ER-positive cancers have been previously described and shown to be important for regulation of ESR1 target genes (PMID: 26169690). To further validate these observations, we used the HRM assay in 7 breast cancer cell lines showing that global DNA methylation is lower in ER-positive cancers (Fig. 5A in the revised manuscript).

• In the ER+/ER- cell lines are there ER methylated/expression differences? Are the target genes differentially methylated pre post SAM-DNMT3A and SAM-DNMT3A-inactive and no sgRNA cells samples?

We did not find any evidence of differential methylation of regions in the different cells we used. It is likely that the difference in sensitivity is because ER+ breast cancer cells are highly dependent on a hypo methylated state.

• Finally, but critically, it is stated that EPICv2 was used for the genome-wide methylation data – however in the methods section outlining the quantification of global methylation- what is described are tools used for EPIVv1 analyses and these do not currently work for EPICv2 as it has a different manifest with a subset of different probes- unless the authors only assessed EPICv1 probes- their analyses need further description.

We thank the reviewer for finding this mistake. For all methylation experiments in this manuscript, we used the EPIC v1.0 array. We have corrected this in the revised manuscript.

Minor

• DNMTs stand for DNA methyltransferases- not DNA methylases as defined in the paper in the abstract and Introduction. Please modify accordingly.

We are sorry for this mistake and have corrected the text accordingly.

• In many parts of the manuscript the writing is 'sloppy', for example In the introduction last paragraph define the SAM system ie the acronym. in the first part of the results 'In this approach a lentiviral vector expressing a catalytically inactive Cas9 enzyme (dCas9) fused to a DNMT enzyme, is stably transduced into a cell'- what DNMT enzymes and what cells?

We are sorry for this and have carefully corrected the manuscript including these examples.